# Room-temperature tetragonal non-collinear Heusler antiferromagnet $Pt_2MnGa$

Sanjay Singh[1], S.W. D'Souza[1], J. Nayak[1], E. Suard[2], L. Chapon[2], A. Senyshyn[3], V. Petricek[4], Y. Skourski[5], M. Nicklas[1], C. Felser[1] & S. Chadov[1]

Antiferromagnetic spintronics is a rapidly growing field, which actively introduces new principles of magnetic storage. Despite that, most applications have been suggested for collinear antiferromagnets. In this study, we consider an alternative mechanism based on long-range helical order, which allows for direct manipulation of the helicity vector. As the helicity of long-range homogeneous spirals is typically fixed by the Dzyaloshinskii–Moriya interactions, bi-stable spirals (left- and right-handed) are rare. Here, we report a non-collinear room-temperature antiferromagnet in the tetragonal Heusler group. Neutron diffraction reveals a long-period helix propagating along its tetragonal axis. *Ab-initio* analysis suggests its pure exchange origin and explains its helical character resulting from a large basal plane magnetocrystalline anisotropy. The actual energy barrier between the left- and right-handed spirals is relatively small and might be easily overcome by magnetic pulse, suggesting $Pt_2MnGa$ as a potential candidate for non-volatile magnetic memory.

[1] Max Planck Institute for Chemical Physics of Solids, Nöthnitzer Str. 40, Dresden D-01187, Germany. [2] Institut Laue-Langevin, BP 156, Grenoble Cedex 9 38042, France. [3] Forschungsneutronenquelle Heinz Maier-Leibnitz FRM-II, Technische Universität München, Lichtenbergstrasse 1, Garching 85747, Germany. [4] Department of Structure Analysis, Institute of Physics ASCR, Na Slovance 2, Praha 18221, Czech Republic. [5] Dresden High Magnetic Field Laboratory (HLD-EMFL), Helmholtz-Zentrum Dresden-Rossendorf, Dresden D-01328, Germany. Correspondence and requests for materials should be addressed to S.S. (email: sanjay.singh@cpfs.mpg.de) or to S.C. (email: stanislav.chadov@cpfs.mpg.de).

Antiferromagnets (AFMs) have attracted increasing attention in state-of-the-art applied and academic research[1–4]. Their auxiliary role of a static support, enhancing the hardness of ferromagnetic electrodes through the exchange-bias effect in conventional microelectronics, has been broadly extended by new perspectives in spintronic applications[1]. For instance, by studying the magnetoresistance effects typically exploited in spintronics[5], it has been demonstrated that an Ir–Mn AFM, utilized as an active medium in a tunnelling magnetoresistance device, exhibits a 160% magnetoresistance at 4 K in weak magnetic fields of 50 mT or less. AFMs also facilitate current-induced switching of their order parameter[6–8] owing to the absence of shape anisotropy and action of spin torques through the entire volume. For example, a relatively low critical current density of 4.6 MA cm$^{-2}$ was reported for the collinear AFM CuMnAs[2]. Additional non-trivial spintronic effects originating from a non-vanishing Berry phase might occur in non-collinear AFMs[9]. Non-collinear planar AFMs without mirror symmetry, such as Mn$_3$Ir, are predicted to exhibit the anomalous Hall[10,11], Kerr, magnetic circular dichroism (MCD) and other effects characterized by the same spatial tensor shape[12], which were not encountered in the AFM systems so far.

Another set of specific properties, alternative to the above-mentioned, are provided by the systems with one-dimensional long-range AFM modulations, such as cycloidal $\mathbf{q} \perp (\mathbf{e}_i \times \mathbf{e}_j)$ and screw (helical), $\mathbf{q} \| (\mathbf{e}_i \times \mathbf{e}_j)$, with $\mathbf{e}_{i,j}$ being the spin directions on $i$ and $j$ neighbouring atomic sites sitting along the spiral propagation vector $\mathbf{q}$. These systems possess a specific order parameter $\boldsymbol{\kappa}_{ij} = \mathbf{e}_i \times \mathbf{e}_j$, denoted as chirality or helicity. For example, in cycloidal AFM insulators, $\boldsymbol{\kappa}$ is coupled to the polarization vector $\mathbf{P} \sim \mathbf{q} \times \boldsymbol{\kappa}$, leading to the first-order ferroelectric effect[13–16]. For the screw-spiral order ($\mathbf{q} \times \boldsymbol{\kappa} = 0$) it becomes possible only upon satisfaction of additional specific conditions, namely, the crystal structure remains invariant under inversion and rotation around $\boldsymbol{\kappa}$, but non-invariant under 180° rotation of the $\boldsymbol{\kappa}$-axis[17,18]. Information transfer in cycloidal spirals along the one-dimensional atomic chains with fixed $\boldsymbol{\kappa}$ stabilized by the surface Dzyaloshinskii–Moriya mechanism was demonstrated by switching their phase with an external magnetic field[19]. Such a scheme is inapplicable to screw spirals owing to their energy degeneracy with respect to $\boldsymbol{\kappa}$ reversal, even if they are deposited on a surface[20]. Similar to the situation with the ferroelectric effect, fixing the helicity of a screw spiral would require additional symmetry constraints on the crystal structure[21]. Despite the cycloidal order seeming to be more ubiquitous for applications, the aforementioned degeneracy between the left- and right-handed magnetic screws in crystals with inversion symmetry might be considered as an alternative advantage, since it allows the direct association of a bit of information with the helicity.

Here, we demonstrate such an AFM screw-spiral magnetic order in the tetragonal Pt$_2$MnGa Heusler system, revealed by neutron diffraction experiments. Additional first-principles analysis justifies the non-relativistic exchange origin of a spiral, confirms its experimentally deduced wave vector $\mathbf{q} \approx (0, 0, 1/5)$ in units of $2\pi/c$, and suggests the cause of the screw-type order as a moderate hard axis (tetragonal $c$ axis) magnetocrystalline anisotropy (MCA).

## Results

### Crystal structure and magnetization
There is no complete experimental information on this material in the literature. The single report on Pt$_2$MnGa[22] briefly refers to it as $L2_1$ AFM with $T_N = 75$ K, but no further details are provided. Later, Pt$_2$MnGa was studied ab-initio by assuming ferromagnetic ordering, which

revealed that the tetragonal phase is more stable[23,24]; therein, it was only mentioned[23] that in the Pt$_x$Ni$_{2-x}$MnGa alloy series, the AFM correlations become stronger with increasing $x$. In a more recent ab-initio study[25], the nearest-neighbor AFM order was found to be noticeably higher in energy compared to the ferromagnetic.

To clear the actual crystal and magnetic structure, we prepared a polycrystalline Pt$_2$MnGa sample. The room-temperature crystal structure (Fig. 1) was deduced from the Rietveld refinement of the X-ray diffraction (XRD) data. All Bragg reflections were well indexed by assuming the tetragonal space group $I4/mmm$. The refined lattice parameters are $a = b = 4.0174(7)$ Å, $c = 7.2393(1)$ Å; Mn, Ga and Pt atoms occupy $2a(0, 0, 0)$, $2b(0, 0, 1/2)$ and $4d(0, 1/2, 1/4)$ Wyckoff sites, respectively (see the inset in Fig. 1).

The low-field $M(T)$ curves measured within the zero-field-cooled (ZFC) and field-cooled (FC) cycles are shown in Fig. 2a. The ZFC $M(T)$ exhibits a maximum at $T \approx 65$ K, which is absent in the FC data. This observation, typical for the tetragonal polycrystalline ferrimagnetic Heusler alloys, may result from a random orientation of the anisotropic crystallites[26]. Overall, the amplitude of $M(T)$ is very small in both the ZFC and FC data, even in high fields at low temperatures (Fig. 2b). The high-temperature behaviour is similar in both the ZFC and FC data and indicates magnetic ordering at 350 K. The possible presence of a small ferromagnetic component modifies the $M(T)$ behaviour, which, at ordering temperature, exhibits a shoulder instead of the peak expected for a conventional AFM. Both isotherms $M(H)$ at 300 K and 2 K (Fig. 2c) exhibit a non-saturating (almost linear) increase up to 7 T, similar to antiferromagnetic or paramagnetic materials. Only a narrow field hysteresis (inset of Fig. 2d) indicates a very weak ferromagnetic component at low temperature. To probe the magnetic response in a very high field, we applied magnetic pulses of 60 T amplitude. The corresponding $M(H)$ curves measured at 257 and 1.5 K (Fig. 2d) increase monotonically and do not saturate with increasing field strength. Only a broadened step-like feature with a small hysteresis within $0 < H < 35$ T is observed at 1.5 K

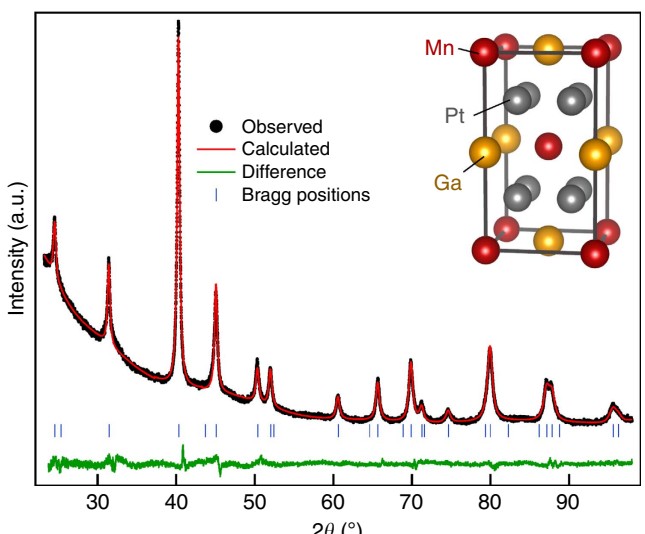

**Figure 1 | Crystal structure of Pt$_2$MnGa.** Rietveld refinement of the room-temperature XRD pattern assuming a tetragonal unit cell with $I4/mmm$ symmetry. Black, red and green lines correspond to the observed, calculated and difference patterns, respectively. The blue ticks indicate the Bragg peak positions. A sketch of the unit cell is shown in the inset: red, yellow and grey spheres indicate Mn, Ga and Pt atoms in $2a$, $2b$ and $4d$ Wyckoff positions, respectively.

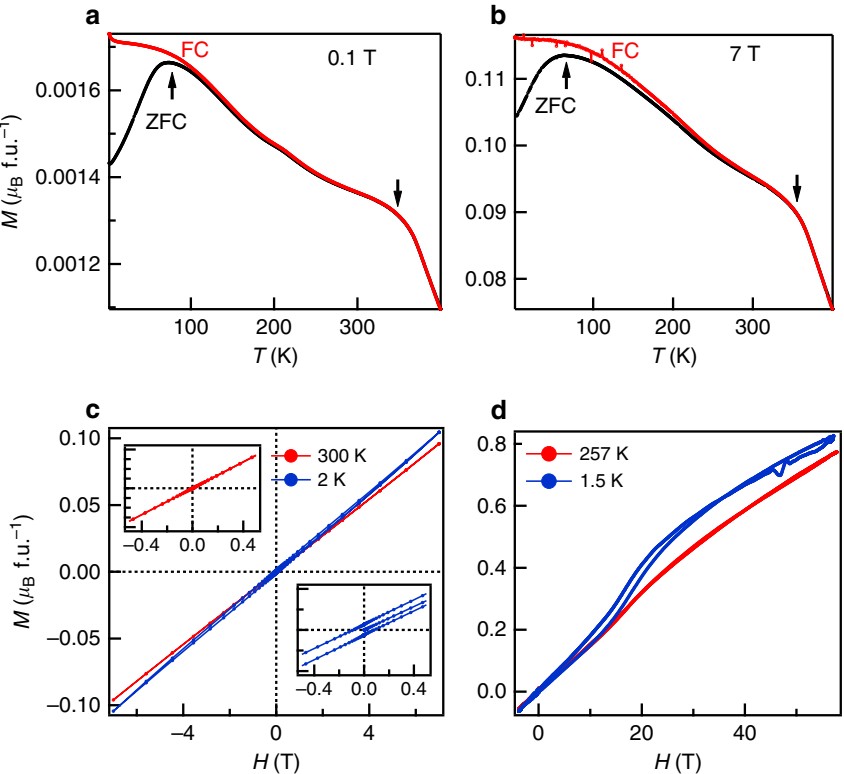

**Figure 2 | Magnetization of Pt₂MnGa.** Temperature-dependent magnetization $M(T)$ (black line—ZFC, red line—FC regime) at (**a**) 0.01 T and (**b**) 7 T. Field-dependent magnetization $M(H)$ hysteresis loops (**c**) up to 7 T at 2 (blue) and 300 K (red) and (**d**) up to 60 T at 1.5 K (blue) and 257 K (red). The insets in (**c**) show the zoomed $M(H)$ behaviour at low magnetic field for 2 (blue) and 300 K (red).

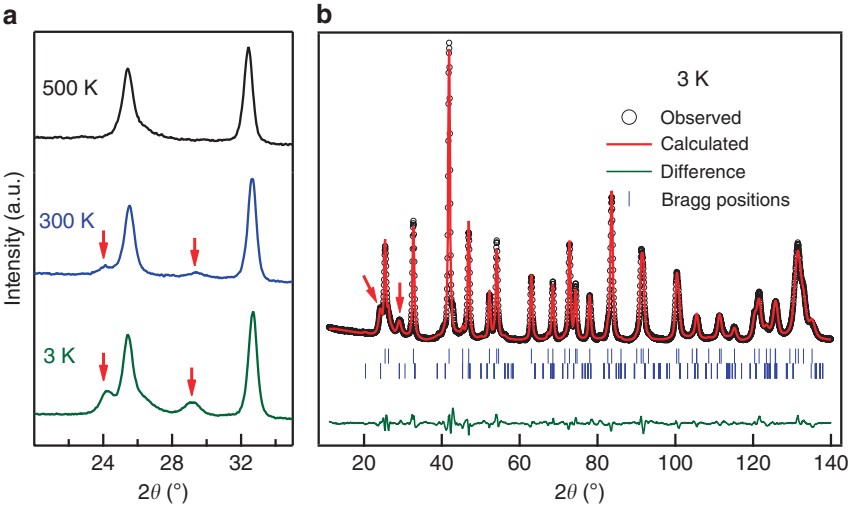

**Figure 3 | Powder neutron diffraction of Pt₂MnGa.** (**a**) Detailed comparison of neutron diffraction patterns at 500, 300, and 3 K. Last two cases exhibit magnetic peaks, which are indicated by red arrows. (**b**) The observed (empty circles), calculated (red line) and difference (green line) neutron diffraction patterns at 3 K within the whole angular range. Upper vertical ticks mark the nuclear peak positions and lower vertical ticks mark the magnetic ones. Red arrows indicate the same magnetic peaks as considered in (**a**).

indicating a metamagnetic transition close to 14 T. This hysteresis might be induced by the non-equilibrium conditions of a magnetic pulse. All these results indicate that Pt₂MnGa is not ferromagnetic.

**Magnetic structure determination using neutron diffraction.** To determine the actual magnetic order, we have measured the powder neutron diffraction at 500 K (above the magnetic ordering), 300 K and 3 K (magnetically ordered phase). At 500 K

it delivers the same crystal structure as room-temperature XRD; however, in addition, due to a higher contrast in scattering amplitudes (0.96, −3.73 and 7.29 fm for Pt, Mn and Ga, respectively), neutrons resolve a certain degree of chemical randomness (see Supplementary Note 1, Supplementary Fig. 1). A comparison of the 500, 300 and 3 K spectra within a narrow angular range ($20° < 2\theta < 35°$) is shown in Fig. 3a. At 300 and 3 K, the long-range magnetic order is clearly evidenced by two additional peaks, at $2\theta \approx 24.1°$ and $29.3°$ (indicated by the red arrows),

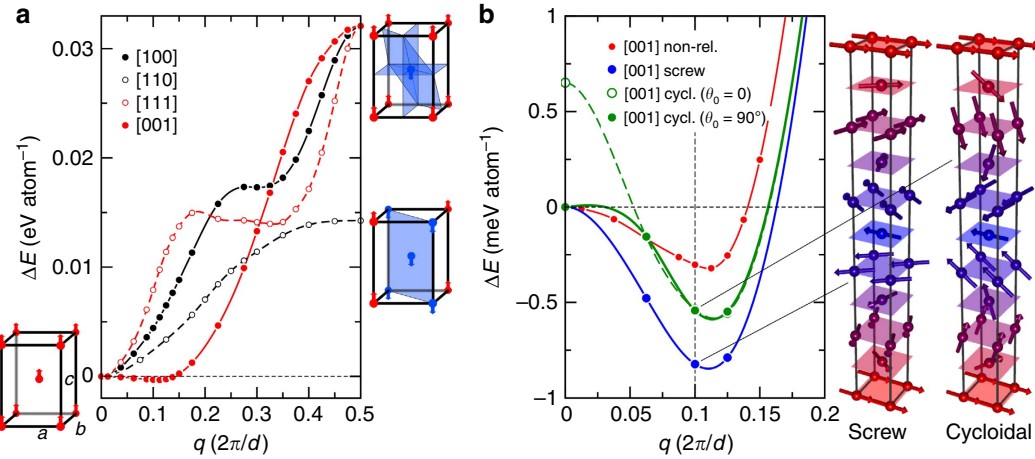

**Figure 4 | Theoretical calculations on Pt$_2$MnGa.** Total energies $\Delta E$ calculated as functions of $q$ (in units of $2\pi/d$, where $d$ is the distance between the nearest Mn-containing planes orthogonal to $q$). (**a**) Non-relativistic regime: black solid and dashed lines refer to [100] ($d = a/2$) and [110] ($d = a/\sqrt{2}$) directions, respectively; red dashed and solid lines—to [111] ($d = (c/2)/\sqrt{2(c/a)^2 + 1}$) and [001] ($d = c/2$) directions, respectively. The energy zero is taken at $q = 0$ (ferromagnet). (**b**) Detailed comparison of different regimes along [001]: red, blue and green lines/points refer to the non-relativistic (same as [001] in (**a**)), screw- and cycloidal-type spirals, respectively. Cycloidal order has two variants: the first one (dashed green line, open circles) at $q = 0$ represents a hard-$c$-axis oriented ferromagnetic state ($\theta_0 = 0$) and the second one (solid green line, filled circles) at $q = 0$ represents an easy-$ab$-plane oriented ferromagnetic state ($\theta_0 = 90°$). The energy zero corresponds to the easy-$ab$-plane ferromagnetic order. All lines are given for an eye guide. For several specific configurations, the Mn($2a$) magnetic sublattice is shown explicitly; spin moments are coloured from red to blue according to their phase.

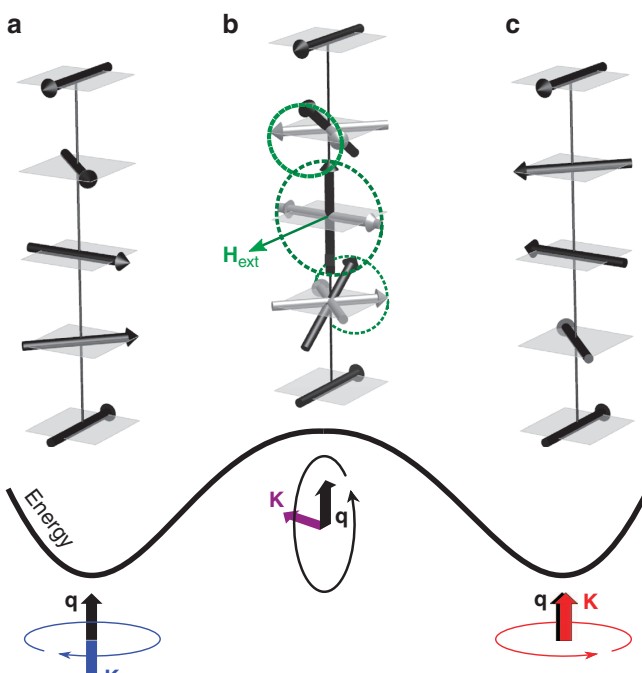

**Figure 5 | Switching of the magnetic helicity using magnetic pulse.** Application of the external magnetic pulse $\mathbf{H}_{ext}$ (indicated by the green arrow) perpendicular to the spiral wave vector $\mathbf{q}$ causes the precession of local moments (rotating along the dashed green curves), which changes the helicity of a spiral from left-handed screw (**a**), through the intermediate cycloidal state (**b**) and to the right-handed screw (**c**). In the case of the easy-plane MCA, the long period of a spiral greatly facilitates reorientation of helicity from $\kappa \uparrow \downarrow \mathbf{q}$ (blue) to $\kappa \uparrow \uparrow \mathbf{q}$ (red) or vice versa, since the top of the energy barrier between two stable screws (**a**,**c**) is a cycloid (**b**) $\kappa \perp \mathbf{q}$ (magenta), in which only few atomic planes with magnetic moments orthogonal to $\mathbf{H}_{ext}$ acquire high energy.

that closely correspond to the commensurate modulation vector $\mathbf{q} = (0, 0, 1/5)$ in units of $2\pi/c$. Detailed analysis reveals slightly incommensurate temperature-dependent variation: $\mathbf{q} = (0, 0, 0.207(2) \cdot 2\pi/c)$ at 300 K and $(0, 0, 0.195(1) \cdot 2\pi/c)$ at 3 K.

To specify more details of the magnetic ground state, we have focused on 3 K data corresponding to the strongest magnetic peak intensities (Fig. 3b). The refinement suggests that the preferable magnetic order is a spiral (in both $2a$ and $2b$ Mn sublattices), which delivers the amplitudes of local moments close to a reasonable value for Mn in these positions ($\sim 4\ \mu_B$), whereas other variants, for example, collinear spin-wave, lead to values substantially exceeding 5 $\mu_B$. In the next step, we tried to distinguish which type of spin spiral is more preferable. By assuming a spiral magnetic structure rotating in the $bc$ plane (cycloid), we obtained the moments of 4.33(13) $\mu_B$ for Mn in both the $2a$ and $2b$ sites. Although these values are reasonable, they exceed those reported in the literature for Mn($2a$) in Mn-based Heusler alloys. Finally, assuming the spiral rotating within the $ab$-plane (screw) leads to 3.93(11) $\mu_B$, a value that is closer to those reported for similar structures[26].

**Theory.** First, we determined the preferential $\mathbf{q}$ vector based on total energy calculations. Because in our case the basic features of a spiral are mostly controlled by the interplay of the isotropic exchange interactions, which have the largest energy scale (Dzyaloshinskii–Moriya interactions are cancelled by symmetry), it is practical to start with the non-relativistic calculations. The absence of spin–orbit coupling allows to apply the generalized Bloch theorem and study the spin spirals in the chemical unit cell without considering large supercells. The energy dispersion computed along several symmetric directions is shown in Fig. 4a. In order to plot several curves along the same coordinate axis, we use the $q$ length in $2\pi/d$, where $d$ is the distance between the nearest Mn-containing atomic planes orthogonal to $\mathbf{q}$. In this notation, $q = 0.5$ corresponds to the antiparallel orientation of the nearest planes. As we see from Fig. 4a, the energy dispersion in

the $ab$-plane ([100] and [110] directions) is monotonous, being characterized by a single minimum at $q = 0$ (ferromagnet) and a single maximum at $q = 0.5$ (shortest AFM order). For the out-of-$ab$-plane directions, one observes a formation of the local minimum within $0 < q < 0.5$. Whereas in the direction [111] it corresponds to rather high energy, in [001], (along the $c$ axis) it becomes global, supporting experimental conclusions. The minimum energy vector appears to be close to the experimental $\mathbf{q} = (0, 0, 0.11) \approx (0, 0, 1/5 \cdot 2\pi/c)$ (see also a more detailed plot in Fig. 4b). A complementary Monte-Carlo simulation of the classical Heisenberg model parameterized by the $ab$-$initio$ exchange coupling constants (see Supplementary Note 2, Supplementary Fig. 2) reasonably reproduces the Neel temperature ($T_N \approx 350$ K) and reveals that the magnetic order is set by the interplay between the strong short-range parallel and the weaker long-range (seventh shell within the Mn($2a$) sublattice) antiparallel interactions along the $c$ axis.

Next, we figured out the role of the relativistic effects, which determines the type of the spin spiral (screw or cycloidal). In this case, the Bloch theorem does not hold and the magnetic order can be studied only in supercells. As we can account only for commensurate modulations, the supercells must be sufficiently large to provide energies at long wavelengths, for example, a minimal supercell hosting a $(0, 0, 1/5 \cdot 2\pi/c)$ modulation contains at least five standard units. As we see from Fig. 4b, the relativistic effects substantially deepen the spiral energy minimum, but almost no change in its $q$ position was observed. The $ab$-plane appears to be an easy plane as the screw-type spiral has the lowest energy in the whole range of the wavelengths. The energy difference between the cycloidal and screw spirals is contributed by the MCA energy, which has a rather large magnitude for the Heusler class (as it follows from the comparison at $q = 0$), being close to 0.65 meV atom$^{-1}$ (or 2.6 meV f.u. $^{-1}$). In the global minimum, this energy difference is reduced more than twice. Owing to the hard-$c$-axis MCA, the cycloidal spiral might be still better optimized in terms of its homogeneity (as it is distorted by the MCA); however, its energy must in any case be higher compared to the screw order. At the same time, as it follows from the energy difference between the cycloidal $\theta_0 = 0°$ and 90° variants, which grows towards smaller $q$, phase optimization makes sense only for $q < 0.06$, that is, far from the global minimum. The amplitude of the Mn moment in the screw spiral has a tendency to grow from $q = 0$ (ferromagnetic) towards $q = 0.5$ (shortest AFM order), though its absolute amplitude increases only slightly: from $\sim 3.7$ to 3.8 $\mu_B$, which agrees with neutron data refinement.

## Discussion

To conclude, we present a tetragonal stoichiometric ($I4/mmm$) Heusler compound, $Pt_2MnGa$, exhibiting room-temperature long-range AFM spiral order, which possesses a unique combination of properties for the Heusler family of materials. $Ab$-$initio$ calculations reasonably reproduce the magnetic modulation vector deduced from neutron diffraction and suggest the origin of the spiral as an interplay of the isotropic exchange interactions within the Mn sublattice. Monte-Carlo simulations of the Heisenberg model parameterized with $ab$-$initio$ exchange coupling constants reasonably reproduce the Neel temperature and suggest the long-range antiparallel Mn–Mn exchange (beyond the sixth Mn shell) as a driving mechanism for the AFM order. Relativistic calculations indicate an easy-$ab$-plane MCA, which stabilizes the screw (proper screw or fully helical) spiral type. Owing to the inversion symmetry, the left- and right-handed spirals are equally stable. In spite of a large MCA, the energy barrier between left- and right-handed spirals can be efficiently overcome via precessional reorientation of magnetization, induced by magnetic pulses perpendicular to the spiral axis (see Fig. 5). Owing to this, the actual barrier reduces to the energy difference between the screw and cycloidal spiral orders and greatly facilitates the reorientation. In particular, this suggests $Pt_2MnGa$ as a convenient candidate for non-volatile magnetic memory based on the helicity vector as a bit of information.

## Methods

**Sample preparation.** The polycrystalline ingot of $Pt_2MnGa$ was prepared by melting appropriate quantities of Pt, Mn and Ga of 99.99% purity in an arc furnace under Ar atmosphere. Ingots were annealed for five days at 1273 K and then quenched in ice water.

**X-ray diffraction.** Crystalline quality was verified at room temperature using powder X-ray diffraction with Cu K$\alpha$ radiation.

**Compositional analysis.** The chemical composition was justified using energy dispersive analysis of X-rays, which delivers $Pt_{50.98}Mn_{25.05}Ga_{23.98}$ ($Pt_{2.04}Mn_{1.0}Ga_{0.96}$); in the manuscript we refer it as $Pt_2MnGa$.

**Magnetic measurements.** The temperature- ($M(T)$) and field-dependent ($M(H)$) magnetization was measured using a SQUID-VSM magnetometer. The low-field $M(T)$ curves were measured within the ZFC and FC cycles. For the ZFC measurement, the sample was cooled in zero field down to 2 K; then, 0.1 T field was applied, and the data were recorded in the heating cycle up to 400 K. Subsequently, the data were recorded in the same field (0.1 T) by cooling from 400 K down to 2 K (FC). The magnetic isotherms $M(H)$ have been measured at 1.2 K and 257 K in pulsed fields up to 60 T (Dresden High Magnetic Field Laboratory).

**Neutron diffraction.** Neutron diffraction measurements were performed at the D2B high-resolution neutron powder diffractometer (ILL, Grenoble). A vanadium cylinder was used as a sample holder. The neutron data were collected at 500 K, 300 K and 3 K using a wavelength of 1.59 Å. The refinement was done using the Fullprof software package[27].

**Computational details.** To complement our experimental results, we computed the $q$-dependent total energies using the $ab$-$initio$ linearized muffin-tin orbitals (LMTO) method, adopting the local spin-density approximation to the exchange-correlation[28]. As the method works for perfectly ordered systems, we have neglected the Mn/Ga chemical disorder indicated by neutron scattering. The unit cell parameters were taken from the present experimental refinement.

**Data availability.** The data that support the findings of this study are available upon request from S.S. and S.C.

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

## Acknowledgements

S.S. thanks Alexander von Humboldt foundation for fellowship. We thank A.N. Yaresko (MPI-FKF Stuttgart) for providing his programme code and A. Chakrabarti and S.R. Barman for useful discussions. The work was financially supported by the ERC AG 291472 'IDEA Heusler!' and HeuMem project within the European network M-era.net. We acknowledge the support of the HLD at HZDR, member of the European Magnetic Field Laboratory (EMFL). V.P. acknowledges the Ministry of Education, Youth and Sports National sustainability programme of Czech Republic (Project no. LO1603).

## Author contributions

C.F. supervised the project. S.S. and J.N. prepared the sample. S.S. performed structural and magnetic measurements and analysis. M.N. planned the high-magnetic-field experiment. S.S. and Y.S. performed the high-magnetic-field experiment. S.S. and E.S. performed the neutron diffraction experiment. S.S. analysed the neutron diffraction data with the help of L.C., A.S. and V.P. S.C. did the spin-spiral calculations. S.W.D. conducted the Monte-Carlo simulations. S.C. and S.S. have written the manuscript with substantial feedback from all co-authors.

## Additional information

**Competing financial interests:** The authors declare no competing financial interests.

