## [Peer Review File · Nature Communications]

REVIEWERS' COMMENTS:

Reviewer #1 (Remarks to the Author):

The research group working on Heusler compounds in the Max-Planck Institute of Physical Chemistry in Dresden is responsible for some of the most recent fascinating results on these compounds. The present manuscript focus on Pt₂MnGa compound. Only little has been known previously on its properties: mainly that it crystallizes in a tetragonal structure and that it seems to adopt an antiferromagnetic structure. The research carried out by the authors reveals that in reality this compound is by far much more complex adopting a helical magnetic spiral structure leading to a long-range antiferromagnetic structure. Such antiferromagnets are important for nowadays spintronics/magneto-electronics and to this respect Pt₂MnGa is a potential materials for non-volatile magnetic memories. The obtained experimental data have been explained using ab-initio calculations which show that the magnetic order is of pure exchange character and that the helicity is fixed by a large basal plane magnetocrystalline anisotropy. To this respect the research carried by authors is novel and is an important advancement in spintronics research.

Both experimental and simulation methods employed are state of the art. Figures/references are adequate and conclusions are well-supported. Taking into account all the above points, I recommend publication of the manuscript with no revision.

Reviewer #2 (Remarks to the Author):

The manuscript by Singh et al reports on the compound Pt₂MnGa, which they show to be a chiral antiferromagnet (AF). Additionally they show that it is a promising candidate for antiferromagnetic spintronics. I found the work to be very interesting and timely. The claims are well supported by data and I recommend it for publication in Nature Communications with a few corrections/additions detailed below:

1) The language needs some work. Even the first two sentences of the abstract contain numerous grammatical errors and perhaps more importantly parts are just difficult to understand. For example, the second sentence refers to "related schemes" which is a very nondescript and ill-defined phrase. There are numerous cases of this throughout the text. The manuscript would benefit greatly from a little polishing.

2)The references to background works are a little scarce. Notable exceptions are the theory paper by Zelezny et al Phys. Rev. Lett. 113, 157201 (2014) on switching antiferromagnetic (AF) systems, but also similar papers to this manuscript on new AF systems published in nat. commun. which would seem natural to reference: Barthem et al Nat. commun. 4, 2892 (2013), Wadley et al Nat. Commun. 4, 2322 (2013)

3) On page 4 the Pt₂MnGa system is discussed. Pt₂MnGa studied in Ref 20 has a lattice parameter which does not match the material studied here, so perhaps the growth conditions are vital in stabilising this form?

4)On page 5, where the magnetometry results are discussed, the maximum at 65K in the ZFC data (fig 3a) is referred to as typical of polycrystalline Heusler alloys. References would be useful here. It would also be nice to have more explanation of why the data differs from that of a canonical AF, where a peak in the susceptibility marks the Neel temperature. Is it on the back of a ferromagnetic component?

In conclusion I think the manuscript unveils a very interesting AF system and its properties are well supported by the data and rigorous analysis. This is a timely study and should be published after a little further work on the language.

Here we would like to thank both Referees for their interest to our study and recommendation for publication. Please, find our replies to the Referees minor concerns.

Sincerely,

The Authors

Reviewer #1 (Remarks to the Author):

The research group working on Heusler compounds in the Max-Planck Institute of Physical Chemistry in Dresden is responsible for some of the most recent fascinating results on these compounds. The present manuscript focus on Pt₂MnGa compound. Only little has been known previously on its properties: mainly that it crystallizes in a tetragonal structure and that it seems to adopt an antiferromagnetic structure. The research carried out by the authors reveals that in reality this compound is by far much more complex adopting a helical magnetic spiral structure leading to a long-range antiferromagnetic structure. Such antiferromagnets are important for nowadays spintronics/magneto-electronics and to this respect Pt₂MnGa is a potential materials for non-volatile magnetic memories. The obtained experimental data have been explained using ab-initio calculations which show that the magnetic order is of pure exchange character and that the helicity is fixed by a large basal plane magnetocrystalline anisotropy. To this respect the research carried by authors is novel and is an important advancement in spintronics research. Both experimental and simulation methods employed are state of the art. Figures/references are adequate and conclusions are well-supported. Taking into account all the above points, I recommend publication of the manuscript with no revision.

We are grateful to the Referee for his inspiring comments and positive evaluation of this study.

Reviewer #2 (Remarks to the Author):

The manuscript by Singh et al reports on the compound Pt₂MnGa, which they show to be a chiral antiferromagnet (AF). Additionally they show that it is a promising candidate for antiferromagnetic spintronics. I found the work to be very interesting and timely. The claims are well supported by data and I recommend it for publication in Nature Communications with a few corrections/additions detailed below:

We appreciate the Referee's positive comments. Below we provide the details on the changes made according to the Referee's suggestion.

1) The language needs some work. Even the first two sentences of the abstract contain numerous grammatical errors and perhaps more importantly parts are just difficult to understand. For example, the second sentence refers to "related schemes" which is a very nondescript and ill-defined phrase. There are numerous cases of this throughout the text. The manuscript would benefit greatly from a little polishing.

We followed the Referee's suggestion by involving native speakers to improve the language throughout the manuscript.

2)The references to background works are a little scarce. Notable exceptions are the theory paper by Zelezny et al Phys. Rev. Lett. 113, 157201 (2014) on switching antiferromagnetic (AF) systems, but also similar papers to this manuscript on new AF systems published in nat. commun. which would seem natural to reference: Barthem et al Nat. Commun. 4, 2892 (2013), Wadley et al Nat. Commun. 4, 2322 (2013)

We thank the Referee for this suggestion. We have added both Mn₂Au and CuMnAs studies to our introductory overview (Refs. [3] and [4]) in the context of the systems with the collinear AFM order.

3) On page 4 the Pt₂MnGa system is discussed. Pt₂MnGa studied in Ref 20 has a lattice parameter which does not match the material studied here, so perhaps the growth conditions are vital in stabilising this form?

It seems that the growth conditions in Ref.20 might substantially differ. We would certainly like to discuss them, but, unfortunately, Ref.20 lacks the synthesis description. It is stated only the annealing temperature of 900 C, but no further information. Thus, from our side we could only try to describe all necessary steps needed to reproduce the synthesis of the tetragonal structure that we obtained.

4)On page 5, where the magnetometry results are discussed, the maximum at 65K in the ZFC data (fig 3a) is referred to as typical of polycrystalline Heusler alloys. References would be useful here.

Indeed, more precisely, it is often observed in the tetragonal polycrystalline ferrimagnets (e.g. Mn-rich Heusler systems). Here we added a relatively recent work of Meshcheriakova et al [26] for Mn₂RhSn system.

It would also be nice to have more explanation of why the data differs from that of a canonical AF, where a peak in the susceptibility marks the Neel temperature. Is it on the back of a ferromagnetic component?

This seems to be the most straightforward mechanism: the small ferromagnetic component can be easily induced by a certain amount of disorder. This indication of a small fm component has been also observed in our M(H) curve. We mentioned this now explicitly in the text.

In conclusion I think the manuscript unveils a very interesting AF system and its properties are well supported by the data and rigorous analysis. This is a timely study and should be published after a little further work on the language.

We thank the Referee for his helpful suggestions, useful remarks, and high ranking of our work.